# Facile Synthesis of the Amorphous Carbon Coated Fe-N-C Nanocatalyst with Efficient Activity for Oxygen Reduction Reaction in Acidic and Alkaline Media

**DOI:** 10.3390/ma13204551

**Published:** 2020-10-13

**Authors:** Linglei Jin, Baikang Zhu, Xuesong Wang, Le Zhang, Debin Song, Jian Guo, Hengcong Tao

**Affiliations:** 1School of Petrochemical Engineering & Environment, Zhejiang Ocean University, Zhoushan 316022, Zhejiang, China; hyukria@163.com (L.J.); zszbk@zjou.edu.cn (B.Z.); wmetxeors@163.com (X.W.); sdb18857057903@163.com (D.S.); 025125@zjou.edu.cn (J.G.); 2United National-Local Engineering Laboratory of Oil & Gas Storage and Transportation Technology, Zhejiang Ocean University, Zhoushan 316022, Zhejiang, China; zhangle@zjou.edu.cn

**Keywords:** oxygen reduction reaction, iron, coating, P-123

## Abstract

With the assistance of surfactant, Fe nanoparticles are supported on *g*-C_3_N_4_ nanosheets by a simple one-step calcination strategy. Meanwhile, a layer of amorphous carbon is coated on the surface of Fe nanoparticles during calcination. Transmission electron microscopy (TEM), scanning electron microscope (SEM), X-ray diffraction (XRD), X-ray photoelectron spectroscopy (XPS), and inductively coupled plasma (ICP) were used to characterize the morphology, structure, and composition of the catalysts. By electrochemical evaluate methods, such as linear sweep voltammetry (LSV) and cyclic voltammetry (CV), it can be found that Fe_25_-N-C-800 (calcinated in 800 °C, Fe loading content is 5.35 wt.%) exhibits excellent oxygen reduction reaction (ORR) activity and selectivity. In 0.1 M KOH (potassium hydroxide solution), compared with the 20 wt.% Pt/C, Fe_25_-N-C-800 performs larger onset potential (0.925 V versus the reversible hydrogen electrode (RHE)) and half-wave potential (0.864 V vs. RHE) and limits current density (2.90 mA cm^−2^, at 400 rpm). In 0.1 M HClO_4_, it also exhibits comparable activity. Furthermore, the Fe_25_-N-C-800 displays more excellent stability and methanol tolerance than Pt/C. Therefore, due to convenience synthesis strategy and excellent catalytic activity, the Fe_25_-N-C-800 will adapt to a suitable candidate for non-noble metal ORR catalyst in fuel cells.

## 1. Introduction

As a significant cathode reaction, oxygen reduction reaction (ORR) has received extensive attention in many sustainable energy storage and conversion fields [1,2,3]. Compared with the anode reaction, composed of the hydrogen oxidation reaction, the cathode reaction, composed of the oxygen reduction reaction, has slow reaction rate. Therefore, the development of an oxygen reduction electrode catalyst with high catalytic activity has promising prospects in scientific research and actual production applications. Among all oxygen reduction catalysts, platinum-based catalysts are generally regarded as the best due to their low overpotential, high current density, and the four-electron transfer process in the reaction. Currently, 20 wt.% Pt/C catalysts are used in various commercial fuel cells. However, platinum on earth is limited and expensive. In addition, platinum catalysts also have poor methanol tolerance and insufficient stability, which are the biggest barriers to the commercial application of fuel cells [4].

In order to break through these barriers, non-precious metal catalysts (NPMCs) have been developed over the last few decades [5]. The transition metal-nitrogen-carbon (M-N-C) composite catalyst has been considered to be a candidate catalyst due to its relatively low cost, high catalytic effect, and excellent stability and durability [6,7]. So far, compared with Pt-based catalysts, higher ORR catalytic activity and durability have been demonstrated by the advanced NPMCs in alkaline electrolyte. However, in acidic medium, the performance of NPMCs still needs to be improved [8,9,10]. Thus, it is of practical worth to develop an NPMC with superior ORR catalytic performance and durability no matter whether it is in alkaline or acidic electrolyte [11].

In the 1960s, for the first time, Jasinski et al. demonstrated that cobalt phthalocyanine containing a M-N_4_ structure has certain catalytic activity for ORR in alkaline solution [12]. Since then, various M-N-C catalysts have been studied and reported. Later, Bagotzky et al. found that, after calcining the transition metal coordinate macrocyclic compound in an inert atmosphere, the ORR catalytic activity and durability of the prepared samples were significantly improved [13]. Subsequently, this new method of calcining the precursor in a specific atmosphere began to be applied. Yeager et al. have since shown that more ordinary and cost-efficient small molecular precursors, consisting of transition metal, nitrogen-rich compounds, and carbon carriers, were able to replace expensive macrocyclic compounds in the process of synthesizing M-N-C catalysts [14]. The accurate properties of the active sites of NPMCs in electrocatalysis are not completely understood, and it is impossible to determine whether transition metals act as active sites or merely promote the formation of N-C active sites [15,16]. Notably, iron content, microporous-mesoporous structure, and nitrogen species on the catalyst surface are some of the key factors controlling ORR catalytic performance of Fe-N-C catalysts [17]. Thus, choosing appropriate precursors can adjust the ORR catalytic performance of NPMCs. Meanwhile, a lot of previous work has proved that pyrolysis can actually increase the graphitization of carbon materials, which can efficiently improve the conductivity of materials. However, high temperatures (possibly >1000 °C) may lead to the decomposition of nitrogen atoms in the carbon matrix. Therefore, in general, the ORR activity and durability of NPMCs depend largely on the pyrolysis process and the alternative for precursors.

Herein, we explore a surfactant-assisted method to synthesize the Fe-N-C catalyst supported on a *g*-C_3_N_4_ nanosheet by a simple one-step calcination strategy. Meanwhile, in the calcination process, the surface of iron nanoparticles is covered with amorphous carbon so as to prepare a stable catalyst with effective ORR activity and selectivity in both acidic and alkaline electrolyte. Graphite-like carbon nitride (*g*-C_3_N_4_) is a non-metallic semiconductor polymer material with a graphite structure [18]. It has attracted much attention due to its adjustable electronic structure, excellent chemical stability, and low manufacturing cost [19,20,21]. The *g*-C_3_N_4_ molecule contains a large number of “nitrogen spots” (consisting of six lone pairs of electrons in nitrogen), and these sites become ideal for binding metals [20]. Therefore, as an ideal nitrogen source, *g*-C_3_N_4_ with high nitrogen content can combine with iron ions to form abundant active sites. At the same time, these active sites will be integrated into the carbon carrier. Previous literature has reported that P-123 (a high molecular weight tri-block copolymer (polyethylene oxide-polypropylene oxide-polyethylene oxide (PEO-PPO-PEO))) can be used as a soft template to form a cambium structure, which contributes towards iron dispersion and expands the surface area [22]. However, there are rare reports on NPMCs that use *g*-C_3_N_4_ as the N source as the support and P-123 as the inducer (surfactant), with competitive ORR performance to Pt in the acidic medium. In this work, we focused on the effects of pyrolysis temperature and precursor content of Fe species on the ORR catalytic activity of the prepared catalyst. SEM, TEM, XRD, X-ray photoelectron spectroscopy (XPS), inductively coupled plasma (ICP), and other physical characterization means were adopted to manifest the composition, bulk phase structure, and surface morphology of the prepared catalyst in this work. Electrochemical analysis methods, such as cyclic voltammetry (CV), linear sweep voltammetry (LSV), and amperometric curve (i-t) have been used to detect the electrochemical activity of catalysts in detail. At the same time, combined with the experimental results, the possible active sites of ORR were also discussed.

## 2. Materials and Methods

### 2.1. Reagent and Materials

Pluronic^®^P-123 and platinum grid electrode were derived from Sigma-Aldrich (Merck, Madrid, Spain). Ferric acetylacetonate (Fe(acac)_3_) and KOH was purchased from Aladdin (Shanghai, China). Commercial 20 wt.% Pt/C was bought from Johnson Matthey Fuel Cells Ltd (London, UK). A total of 5 wt.% Nafion solution was derived from DuPont (Wilmington, DE, USA). Other chemicals were bought from Sinopharm Chemical Reagent Co., LTD (Beijing, China). All aqueous solutions used were prepared with deionized water from a Millipore system (Zhoushan, China). All chemical reagents were of analytical grade and used as received without further purification. All potentials in this work were versus the reversible hydrogen electrode (RHE).

### 2.2. Preparation of g-C_3_N_4_

The *g*-C_3_N_4_ were synthesized by using the previous reported methods [18]. Melamine was dried in an oven at 80 °C for 12 h. Dried melamine was then added in an alumina crucible and covered with the crucible lid with tin foil. The tightly wrapped crucible was calcined at 550 °C in the chamber electric furnace for 2 h. Eventually, *g*-C_3_N_4_ was obtained after fully grinding. 

### 2.3. Preparation of Fe_X_-N-C-Y Electrocatalysts

Fe_25_-N-C-800 was taken as a typical example of Fe-N-C catalyst synthesis. First, 300 mg prepared *g*-C_3_N_4_ and 600 mg P-123 were dissolved in 100 mL deionized water and sonicated for 2 h to form a homogenous dispersion. The well dispersed solution was then stirred vigorously by a magnetic stirrer for 2 h. Subsequently, 25 mg Fe(acac)_3_ was sonicated continuously for 2 h after being added into the dispersed solution. After continuous stirring for 12 h, the mixture was dried in an oven at 80 °C for 10 h, and then the dried solids were collected and put into a vacuum tube high-temperature sintering furnace. Under the protection of nitrogen (60 mL·min^−1^), the dried solids were calcined at 550 °C for 2 h and 800 °C for another 2 h. The product of pyrolysis was soaked in pre-prepared 2 M HCl for 24 h, washed with water, and dried to obtain Fe_25_-N-C-800. X in Fe_X_-N-C-Y catalyst represents the amount of iron precursor added (X = 0, 5, 25, 50 mg). Y represents the calcination temperature during the preparation process (Y = 700, 800, 900 °C). All heating rates in this work were 5 °C·min^−1^.

### 2.4. Materials Characterization

The characterization of the catalyst mainly included crystal structure, microscopic morphology, element composition, and pore size distribution. The X-ray diffractometer model DX-2700 (Dandong Fangyuan Instrument Co. LTD, Dandong, China) was used for testing. The X-ray tube was copper palladium (λ = 1.5417 Å), the scanning step width was 0.02°, and the scanning angle was 10°~80°. The tube voltage and tube current were set to 40 kV and 30 mA, respectively. A scanning electron microscope (SEM, Zeiss SIGMA 300 field-emission, Oberkochen, Germany) characterized the microscopic morphology of the catalyst. Transmission electron microscopy (TEM) (FEI Tecnai G2 F20, FEI, Eindhoven, The Netherlands) collected information at a voltage of 200 kV. An inductively coupled plasma optical emission spectrometer (ICP-OES) measurement was performed with Agilent 720ES (Santa Clara, CA, USA), and the actual proportion of iron in the catalyst with different iron precursor content was determined. The K-Alpha type of Thermo Scientific (Waltham, MA, USA) was used to test the catalyst to obtain the valence state and surface energy state distribution of the catalyst by analyzing the X-ray photoelectron spectroscopy (XPS) (Thermo Scientific, Waltham, MA, USA).

### 2.5. Preparation of the Working Electrodes

All electrochemical characterization in this work was measured by a rotating disk electrode (RDE) with an electrode diameter of 5 mm. Before the electrochemical test, the glassy carbon electrode was polished with 0.5 μm and 0.05 μm alumina slurry, respectively. A 3 mg Fe-N-C catalyst was added into a mixed solution of 300 μL ethanol, 300 μL deionized water, and 300 μL 1 wt.% Nafion solution. The mixed solution was sonicated (KQ5200DE, 40 kHz, Kun Shan Ultrasonic Instruments Co., Ltd, Kunshan, China) for 0.5 h and then used as catalyst ink. Then, 7.95 μL catalyst ink was drip-evenly applied to the electrode surface of the RDE. The catalyst load was maintained at 0.135 mg cm^−2^.

### 2.6. Electrocatalytic Measurements

Electrochemical characterization was mainly carried out by CV, LSV, and i-t. Electrochemical characterization of the catalysts were measured by the CHI electrochemical station (CHI 660E, Shanghai Chenhua Instrument Co., Ltd., Shanghai, China) in a standard three-electrode electrochemical cell at room temperature. During the test, a platinum wire electrode was used as the counter electrode, Ag/AgCl electrode as the reference electrode, and glassy carbon electrode loaded with catalyst as the working electrode. The potential was converted to a potential versus RHE according to E: (vs. RHE) = E (vs. Ag/AgCl) + 0.197 + 0.0591 pH.

The ORR catalytic activity of the sample was researched in both alkaline (0.1 M KOH) and acidic (0.1 M HClO_4_) electrolyte. After an experimental device was set up, O_2_ was introduced until the O_2_ in the electrolyte was saturated. All electrochemical measurements were conducted in the O_2_-saturated media. For cyclic voltammetry (CV), 20 CV tests were scanned in the voltage range of 0 to 1.20 V (vs. RHE) at a sweep rate of 100 mV s^−1^. For linear sweep voltammetry (LSV), ORR, active at the as-prepared catalyst, was surveyed by sweeping the catalysts between 1.20 to 0 V (vs. RHE) at 5 mV s^−1^ under 0, 100, 400, 900, and 1600 rpm (increasing from low speed to high speed). In this work, the ORR performance of each catalyst was tested more than four times, and the repeatable data were selected in our results analysis.

The electron transfer number of oxygen molecules on the electrode during the ORR process was evaluated via the rotating disk electrode and following Koutecky–Levich equations.
(1)$$\frac{1}{J}=\frac{1}{J_{L}}+\frac{1}{J_{k}}=\frac{1}{B\omega^{\frac{1}{2}}}+\frac{1}{J_{k}}$$
(2)$$B=0.62nFC_{0}{\left(D_{0}\right)}^{\frac{2}{3}}v^{-\frac{1}{6}}$$
(3)$$J_{k}=nkFC_{0}$$

J, $J_{L}$, and $J_{k}$ are the measured current density, diffusion-limiting current density, and kinetic-limiting current density at a specific potential, respectively. n is the overall number of electrons transferred during the reaction process. F is the Faraday constant (96485 C/mol). $C_{0}$ is the solubility of oxygen in this electrolyte, in 0.1 M KOH, $C_{0}=1.2\times 10^{-6}\;\mathrm{mol}/{\mathrm{cm}}^{3}$; in 0.1 M HClO_4_, $C_{0}=1.26\times 10^{-6}\;\mathrm{mol}/{\mathrm{cm}}^{3}$. v is the kinematic viscosity of the solution, in 0.1 M KOH, $\nu =0.01{\;\mathrm{cm}}^{-2}/s$; in 0.1 M HClO_4_, $\nu =1.009\times 10^{-2}{\;\mathrm{cm}}^{-2}/s$. $D_{0}$ is the diffusion coefficient of O_2_ in the solution, in 0.1 M KOH, $D_{0}=1.9\times 10^{-5}{\mathrm{cm}}^{-2}/s$, in 0.1 M HClO_4_, $D_{0}=1.93\times 10^{-5}{\mathrm{cm}}^{-2}/s$.

## 3. Results

### 3.1. Preparation and Physical Characterization of Fe-N-C

A brief synthesis method of the as-prepared catalyst in this work is illustrated schematically in Figure 1. First, the three precursors, *g*-C_3_N_4_, P-123, and Fe(acac)_3_), were mixed together (details of the experiment are shown in Section 2.3). Subsequently, the thoroughly mixed and dried mixture was pyrolyzed at 800 °C (under nitrogen protection). 

The microscopic morphology of the Fe_25_-N-C-800 was examined by SEM (Figure 2a,b). Combining Figure 2a,b, a mixture of porous fold amorphous carbon and iron nanoparticles is displayed. In Figure 2b, it can be clearly observed that many nano-particle balls are evenly distributed on the carbon folds (bright white dots in Figure 2b), combined with XRD analysis, and it can be seen that the nanoparticles are formed by the agglomeration of iron elemental and Fe_3_C. Subsequently, the Fe_25_-N-C-800 catalyst was subjected to TEM testing (Figure 2c,d), which further confirmed that the catalyst was made of aggregated nanoparticles. The measurement found that the size of the nanoparticle ranged from 5 to 70 nm, and the average size was calculated to be 18.53 nm (Figure 2c, inset). Figure 2d is a high-resolution TEM image of the catalyst. It can be clearly seen that the iron nanoparticles were coated with amorphous carbon. Previous studies have shown that wrapping transition metal compounds in amorphous carbon enables them to interact strongly with each other, leading to synergistic effects that effectively enhance their electrochemical properties [23]. The lattice fringe spacing in Figure 2d was measured to be about 0.204 nm, which was consistent with the (110) crystal plane of elemental iron and the (220) crystal plane of Fe_3_C. This shows that the Fe_25_-N-C-800 catalyst formed elemental iron and Fe_3_C. For further research on the catalytic property of iron element in ORR, the elemental composition of the as-prepared catalyst in this work was evaluated by a high-angle annular dark-field scanning transmission electron microscope (HAADF-STEM). It is shown in Figure 2e,h that many uniformly scattered bright spots were on the atomic scale. We can see that these bright spots were assigned to iron atoms. In this case, they can prove the existence of the iron element [6].

Figure 3a shows the X-ray diffraction pattern of catalysts with a different iron content. The spectra of different samples were roughly similar and there was no significant difference. It can be clearly seen that an amorphous peak of C appeared at 26°. Each sample (except Fe_0_-N-C-800) had a set of apparent peaks at 44.67° and 65.02°. These peaks were consistent with the (110) and (200) crystal planes of Fe (Joint Committee on Powder Diffraction Standards, JCPDS No.06-0696), respectively. Meanwhile, each sample (except Fe_0_-N-C-800) had a set of peaks at 37.74°, 39.79°, 42.87°, 43.74°, 44.56°, 45.86°, and 49.11°. These peaks were for Fe_3_C (JCPDS No. 35- 0772) (210), (002), (211), (102), (220), (112), and (221) crystal planes. This also corresponded to the results obtained from the TEM characterization test. As displayed in Figure 3a, it can be seen that, with the difference of iron precursor content, the characteristic peaks of Fe and Fe_3_C were gradually obvious, from the inconspicuous “broad and relatively weak” peak to the clear “high and narrow” peak. Fe_0_-N-C-800 sample preparation process did not add iron precursor, so it does not contain iron element or iron characteristic peak. Combined with the XRD test data, according to the Scherer formula ($D=\frac{k\lambda }{\beta cos\theta }$), we calculated that the size of the crystal grain of Fe_25_-N-C-800 was 17 nm, which was almost consistent with the data obtained by TEM. Meanwhile, ICP was used to determine the actual proportion of iron in the Fe_X_-N-C-800 catalyst under different iron precursor contents. It is shown in the results indicated in Table 1 that the content of iron was positively correlated with the amount of iron precursor added during the preparation process. That is, the content of iron in the sample increased with the addition of the precursor.

An XPS test was carried out on the Fe_25_-N-C-Y catalyst. The spectrum was a rough sweep of the full spectrum, so the signal of some elements was not obvious. Figure 3b shows the measured spectra of the catalyst compounded at various pyrolysis temperatures. It is shown in Figure 3b that the Fe-N-C samples consisted of carbon, nitrogen, oxygen, and iron. As shown in Table 2, with the increase of calcining temperature, the nitrogen content decreased from 5.22% to 2.48%. Through comparison, it can be clearly seen that the signal intensity of the N element of the sample calcined at 700 °C was more obvious. The possible cause for this phenomenon was that the combination of Fe(acac)_3_ and *g*-C_3_N_4_ ha poor high-temperature resistance under the participation of the surfactant P-123, with which the N element in *g*-C_3_N_4_ was pyrolyzed and volatilized with the rise of pyrolysis temperature. According to previous studies, N-Fe sites are beneficial to increase the ORR catalytic activity of samples [6]. Therefore, it can be reasonably speculated that high pyrolysis temperature is conducive to forming N-Fe sites in the sample, but as the temperature rises, the nitrogen-containing sites will be decomposed at excessively high pyrolysis temperature. In consequence, the Fe_25_-N-C-800 catalyst manifests more excellent ORR performance than catalysts synthesised at other pyrolysis temperatures.

To obtain further information about the valence state of atom and surface energy distribution of the catalyst, XPS tests were carried out on the Fe_25_-N-C-800 catalyst (Figure 3c–e). In this experiment, the XPS spectrum data of all samples were corrected using the carbon–carbon double bond peak at C 1s 284.8 eV. It can be clearly seen in the C 1s XPS binding energy region of the typical sample that the signal was deconvoluted into five peaks (Figure 3c). The peaks at 283.3, 284.2, 285.5, 286.9, and 289.1 eV can correspond to Fe_3_C, C=C, C=N, C-N, and O-C=O, respectively [24,25]. Among them, Fe_3_C had a positive effect on ORR. Figure 3d reveales the N 1s binding energy region of the Fe_25_-N-C-800 catalyst. The signal is deconvolved into three peaks at 397.8, 399.5, and 400.4 eV, belonging to pyridinic N, pyrrolic N/Fe-N_X_, and graphitic N, respectively [5,26,27,28]. At the same time, we calculated the relative composition of the pyridinic N, pyrrolic N/ Fe-N_X_, and graphitic N species (atomic factions) in the samples (Figure 3f). The content of pyridinic N, pyrrolic N/Fe-N_X_, and graphitic N in Fe_25_-N-C-800 catalyst were 38.8%, 36.8%, and 24.4%, respectively. According to previous reports, ORR active sites in M-N-C catalysts may be located around the carbon phase [29]. Meanwhile, previous studies have demonstrated that pyridinic N and pyrrolic N are always located on the graphitic edge [30]. Therefore, pyridinic N and pyrrolic N are significant factors to enhance ORR catalytic reaction. Pyridinic N is a six-member heterocyclic compound, containing one nitrogen heterocyclic atom. Each N atom is bound to two carbon atoms and provides one p-electron to the aromatic π system [31]. Pyrrolic N is a five-member heterocyclic compound containing one azo atom, which binds to two carbon atoms and provides two p-electrons to the π-conjugated system [31]. Previous studies have demonstrated high proportions of pyridinic N and pyrrolic N boost O_2_ reduction by enhancing the current density, spin density, and π state density of C atoms near the Fermi level [32]. Meanwhile, the two-dimensional planar structure of pyridinic N and pyrrolic N can maintain the original planar large π bond structure on the surface of carbon materials and have good electrical conductivity, so they have excellent ORR catalytic activity. However, due to the uneven three-dimensional structure of graphite nitrogen, the original π-conjugated large bond on the carbon surface is destroyed and the conductivity of carbon materials is reduced. Therefore, the catalytic activity of ORR is relatively low. Since nitrogen doping can cause disorder of the structure, a greater number of active sites are provided by the type of N-active site, such as pyridinic N, pyrrolic N, and Fe-N_X_, thereby promoting the reaction process of ORR [31,32,33,34]. Consequently, the Fe_25_-N-C-800 catalyst with the total content of pyridinic N and pyrrolic N/Fe-N_X_ up to 76% showed excellent ORR catalytic performance.

In addition, the signal of the Fe 2p shown in Figure 3e was deconvolved into two main peaks, attributing to Fe 2p_3/2_ and Fe 2p_1/2_, respectively. It can be found that the Fe 2p_3/2_ binding energy region showed various peaks (710.4, 714.0, 719.3 eV), indicating that there were many chemical states of iron species in Fe_25_-N-C-800. Based on previous studies, it can be seen that the peak of Fe 2p_3/2_, at about 710.4 eV, is attributed to Fe-N_X_, which can accelerate O_2_ adsorption and 4e-reduction and has electrocatalytic activity for ORR [11,35,36,37,38]. Meanwhile, the 706.7 eV peak can be attributed to zero-valent iron and Fe_3_C [39]. Consistent with previous reports, the remaining peak at about 723.9 eV is attributed to Fe 2p_1/2_ [6].

Iron is commonly believed to act as a crucial part in enhancing the ORR performance of electrocatalysts. It can improve the catalytic activity of electrocatalysts by promoting the formation of active sites or directly participating in catalyzing ORR [40]. The existence of Fe-N_X_ was confirmed by the XPS spectra of the samples prepared in this work. Based on extensive literature reviews, it was found that Fe-N_4_ is the most effective free active site of metal in oxygen reduction reaction among cathode catalysts of fuel cells [41]. Therefore, in combination with the ORR activity assessment results of the catalyst prepared in this work, and considering that the single iron atom in the metallic state is very active and unstable under electrochemical conditions, we reasonably speculated that the iron atom in Fe_25_-N-C-800 coordinated with the N to form the Fe-N_4_ active site. Meanwhile, because Fe_3_C nanoparticles can enhance the active site in ORR, a sufficient number of Fe_3_C nanoparticles also have a certain impact on the activity of ORR. Considering all the above characteristics, we expect that Fe_25_-N-C-800 catalyst embracing Fe-N_4_ and Fe_3_C can reveal excellent ORR activity.

### 3.2. ORR Catalytic Activity Evaluation

As a series of previous studies revealed, pyrolysis temperature greatly affects the ORR catalytic performance of Fe-N-C catalyst [42]. To explore the impression of pyrolysis temperature and obtain the suitable temperature, the ORR performance of the Fe_25_-N-C catalyst at various temperatures (in the range of 700–900 °C) was investigated. ORR performance was tested in 0.1 M KOH, and the consequences proved that Fe_25_-N-C-800 exhibited the excellent ORR performance with larger onset potential and half-wave potential (Figure 4a). This may be due to the fact that the conductivity, porosity, and active site density of the catalyst prepared at the pyrolysis temperature of 800 °C reached an optimal balance.

Pt/C catalyst is recognized as an effective catalyst for commercial fuel cell cathode reaction (ORR). Therefore, researchers compared ORR catalytic activity evaluation with platinum-based catalysts [4]. The electrocatalytic activity of the as-prepared Fe-N-C catalyst for ORR was evaluated using RDE voltammetry and confronted with the Pt/C catalyst in this work. Previous studies have put forward diverse assumptions for active sites in Fe−N-C catalysts that can enhance ORR properties. In general, there are two main types: one is doped with carbon-based N atoms [11,43,44] and the other is the active site formed by iron species, including Fe-N_X_, Fe_3_C, etc. [45,46,47]. Figure 4b provides the LSV curves of the sample with various iron precursor content. It can be found that the onset potential of the sample with the iron precursor (0.887–0.925 V vs. RHE) was significantly better than the sample without the iron precursor (0.794 V vs. RHE), which also facilitated faster reaction kinetics and a higher electron transfer number. Therefore, we can confirm that the active site of iron speciation serves as a significant part in the process of promoting ORR electrocatalytic properties.

It is shown in Figure 5a that the onset potential of Fe_25_-N-C-800 (0.925 V vs. RHE) was mildly more negative than Pt/C (0.934 V vs. RHE), while its half-wave potential (0.864 V vs. RHE) was slightly more positive than Pt/C (0.858 V vs. RHE). Figure 5b displays the polarization curves of the Fe_25_-N-C-800 catalyst at various rotation rates (0~1600 rpm). With the enhancement of the rotating speed, the diffusion rate of O_2_ in the electrolyte gradually accelerated, which lead to the gradual increase of the limiting of current density of the catalyst.

According to the RDE dates of Fe_25_-N-C-800 at various rotating speeds, combined with the K-L equations, the overall number of electrons transferred during the oxygen reduction process can be calculated. The K-L curves of Fe_25_-N-C-800 reveal an obvious linear relationship, indicating that the reaction process is a first-order kinetic process [48]. After calculation, the average electron transfer number (n) of the oxygen molecules of Fe_25_-N-C-800, over the potential range from 0.2–0.7 V, was 3.86 (Figure 5c), which was approximate to 20 wt.% Pt/C (n = 3.93). Furthermore, as shown in Figure 5d, the Tafel slope is considered to be another crucial parameter for investigating ORR catalytic activity, which mainly investigates the reaction rate of the reaction. In 0.1 M KOH, Fe_25_-N-C-800 had the lowest Tafel slope (29 mV per decade) and lower overpotential compared with other iron precursor content catalysts and Pt/C catalysts.

Previous researchers have found that the Fe-N-C catalyst has a stable structure but unstable electrochemical performance when exposed to H_2_O_2_ (ORR by-product) in acidic electrolyte [49]. The catalytic active site of the Fe-N-C catalyst was not be affected, but its conversion frequency was reduced through the oxidation of the carbon surface, resulting in the weakening of the binding ability of the Fe-N-C catalyst to O_2_. Therefore, compared with the alkaline electrolyte, the ORR performance and durability of the Fe-based catalyst in the acidic electrolyte was significantly reduced. It is noteworthy that the Fe_25_-N-C-800 catalyst also had a certain performance for ORR in the harsh acidic medium. Compared with commercial 20 wt.% Pt/C, although the Fe_25_-N-C-800 catalyst’s onset potential and half-wave potential showed a certain degree of negative shift, its limiting current density was much bigger in 0.1 M HClO_4_ (Figure 6a). Figure 6b displays the LSV curves of the Fe_25_-N-C-800 catalyst at various rates (0~1600 rpm). We can see that, as the rotation speed increased, the O_2_ diffusion rate increased and the limiting current density of the catalyst gradually increased, which was the same as when the electrolyte was alkaline. Meanwhile, the K-L plots of the Fe_25_-N-C-800 catalyst presented an obvious linear relationship. Over the potential range from 0.1–0.4 V, the average number of electrons transferred from the oxygen molecules of Fe_25_-N-C-800 during the reaction process was 3.79 (Figure 6c), which was close to that of Pt/C (n = 3.85). This indicates that, even in harsh acidic solutions, there were nearly four electron transfer ORR pathways. It is shown in Figure 6d that the Tafel slopes of the Fe_25_-N-C-800 catalyst and 20 Wt. % Pt/C were approximately 34 and 45 mV per decade, respectively. The results demonstrated that, in the low overpotential region, Fe_25_-N-C-800 also had a similar oxygen reduction reaction mechanism to Pt/C.

### 3.3. ORR Durability Characterization

In practical applications, catalyst stability and methanol tolerance are also important parameters for direct methanol fuel cells. As shown in Figure 7a,c, either in a 0.1 M KOH or 0.1 M HClO_4_ media, after adding CH_3_OH (3 M), the current response of Pt/C dropped sharply, while the current response of Fe_25_-N-C-800 did not change significantly. The consequences prove that Fe_25_-N-C-800 takes on prominent resistance to the methanol crossover effect. 

Figure 7b,d illustrates the electrocatalytic durability of the Fe_25_-N-C-800 catalyst and Pt/C catalyst. After 28,800 s chronocurrent electrolysis in 0.1 M KOH media, the current density of the Fe_25_-N-C-800 catalyst dropped to 91% of the initial current, whereas the current density of the Pt/C catalyst was only 82%. In 0.1 M HClO_4_ media, the current density of the Fe_25_-N-C-800 catalyst left about 83% of the initial current, while that of Pt/C sharply dropped, leaving only 71% of the initial current. According to previous studies, this is because Pt may dissolve in the electrolyte, aggregate into larger particles, and separate from the carrier, resulting in poor durability in acidic media [50].

Table 3 summarizes the comparison of ORR performance between Fe_25_-N-C-800 prepared in this work and the recently reported Fe-N-C catalyst in alkaline and acidic electrolyte. The electron transfer number of Fe_25_-N-C-800 at different potentials was close to 4. Therefore, the oxygen reduction reaction on the Fe_25_-N-C-800 catalyst followed the efficient four-electron path, that is, the oxygen was completely reduced to water, which proves that the catalyst of our vegetation had effective ORR catalytic activity. At the same time, Table 3 confirms that the durability of the catalyst prepared in this work was superior to other types of Fe-N-C to a certain extent, no matter whether it was under alkaline or acidic electrolyte.

The excellent stability and durability of the Fe_25_-N-C-800 catalyst perhaps contributed to the fact that Fe nanoparticles were wrapped by amorphous carbon to protect metals from being dissolved. A similar result was observed in previous reports, that is, the encapsulated iron nanoparticles cannot be dissolved in acid [56]. A series of results show that the Fe_25_-N-C-800 catalyst possesses a certain prospect in the application of fuel cells, according to its satisfactory methanol resistance and durability in both alkaline and acidic media.

## 4. Conclusions

To sum up, we used surfactants to support Fe nanoparticles on *g*-C_3_N_4_ nanosheets through one-step calcination. Meanwhile, a layer of amorphous carbon was formed on the surface of Fe nanoparticles, thereby preparing an iron-based nanocomposite catalyst. According to a series of physical characteristics, such as SEM, TEM, XRD, and XPS, we proposed that the iron atoms, Fe_3_C nanoparticles, and Fe-N_X_ active sites produced during the pyrolysis process raised the ORR catalytic activity to some extent. The optimal pyrolysis temperature (800 °C) and iron precursor content (25 mg) were determined by preparing various catalysts. The Fe_25_-N-C-800 catalyst exhibited comparable catalytic activity to commercial 20 wt.% Pt/C in alkaline electrolyte. Via a series of electrochemical evaluations, it was detected that the initial potential of the Fe_25_-N-C-800 was 0.925 V vs. RHE, the half-wave potential was 0.864 V vs. RHE, and the number of electron transfers was 3.86. At the same time, its excellent stability and methanol resistance were manifested because of the amorphous carbon protection. In acidic electrolyte, the Fe_25_-N-C-800 catalyst also had a four-electron transfer ORR pathway and, at the same time, exhibits more excellent methanol tolerance and stability than Pt/C. Therefore, the catalyst prepared for this work can be used as a fuel cell cathode catalyst.

## Figures and Tables

**Figure 1 materials-13-04551-f001:**
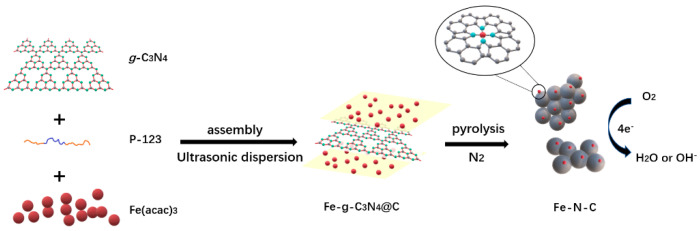
Schematic synthetic strategy.

**Figure 2 materials-13-04551-f002:**
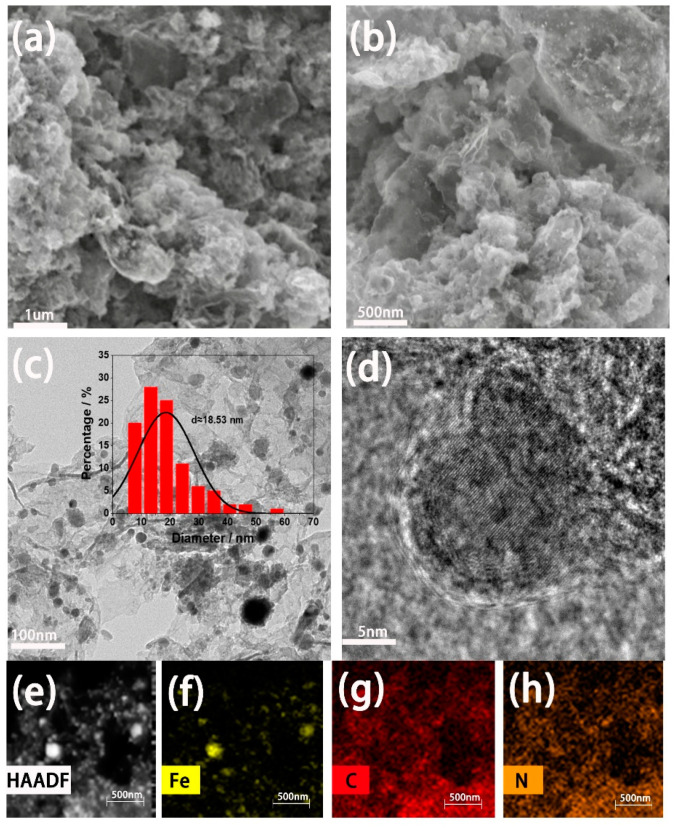
Characterizations of the Fe_25_-N-C-800 catalyst. (**a**,**b**) SEM images at different magnifications; (**b–d**) TEM images at different magnifications; (**e–h**) elemental-mapping images of high-angle annular dark-field (HAADF), iron, carbon, and nitrogen.

**Figure 3 materials-13-04551-f003:**
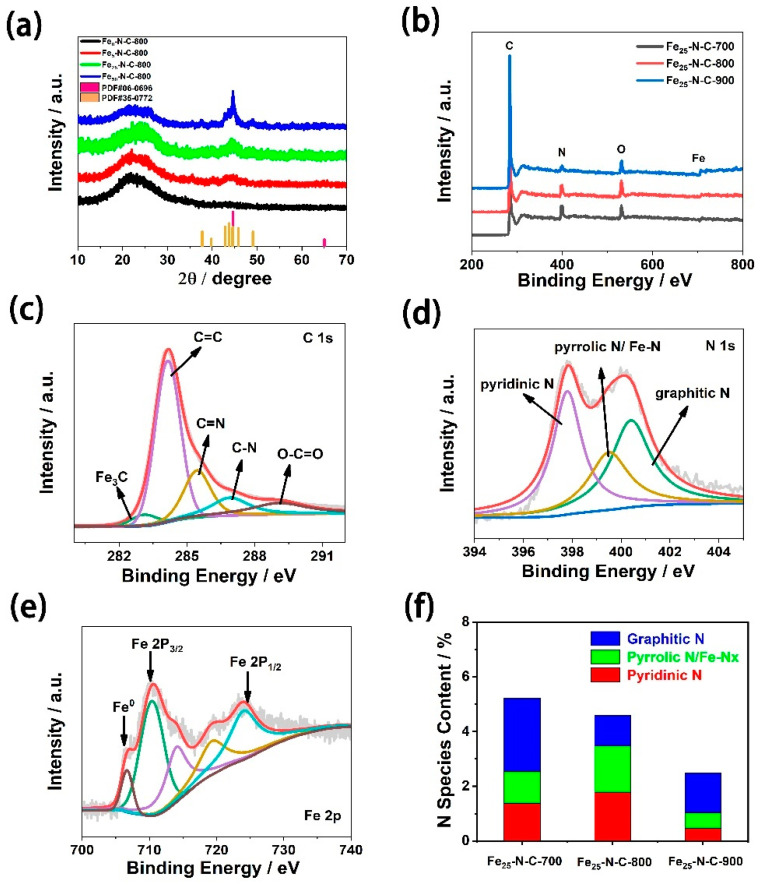
(**a**) XRD of Fe_X_-N-C-800 (X = 0, 5, 25, 50 mg); (**b**) XPS spectra of Fe_25_-N-C-Y (Y = 700, 800, 900 °C); (**c**) XPS C 1s spectra of Fe_25_-N-C-800; (**d**) XPS N 1s spectra of Fe_25_-N-C-800; (**e**) XPS Fe 2p spectra of Fe_25_-N-C-800; (**f**) the relative contributions of the various N configurations estimated from the N 1s spectra of Fe_25_-N-C-Y catalysts (Y = 700, 800, 900 °C).

**Figure 4 materials-13-04551-f004:**
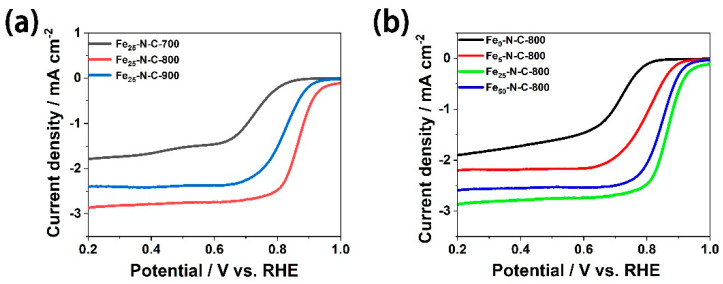
The electrocatalytic behavior in O_2_-saturated 0.1 M KOH (potassium hydroxide solution) at a scan rate of 10 mV s ^−1^ and a rotation rate of 400 rpm. (**a**) LSVs of Fe_25_-N-C-Y (Y = 700, 800, 900 °C); (**b**) LSVs of Fe_X_-N-C-800 (X = 0, 5, 25, 50 mg).

**Figure 5 materials-13-04551-f005:**
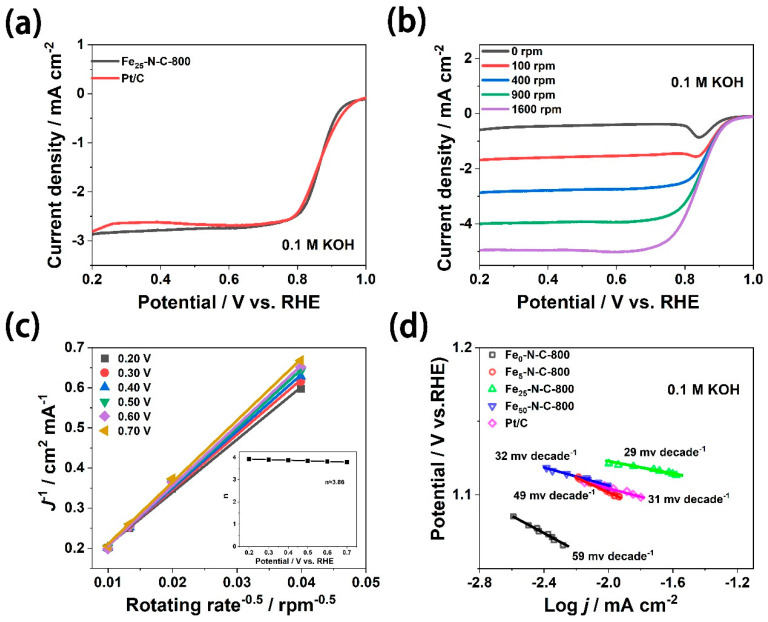
Characterizations of oxygen reduction reaction (ORR) activity in alkaline medium. (**a**) The rotating disk electrode (RDE) dates of Fe25-N-C-800 and Pt/C, rotation rate = 400 rpm. ((**b**) Polarization curves of Fe25-N-C-800 at various rotation rates (increasing from top to bottom); (**c**) Koutecky–Levich plots of Fe25-N-C-800 and number of electrons transferred (inset) at various potentials (0.2 –0.7 V vs. RHE); (**d**) Tafel slope plots of Pt/C and FeX-N-C-800 (X = 0, 5, 25, 50 mg) derived from the LSV results.

**Figure 6 materials-13-04551-f006:**
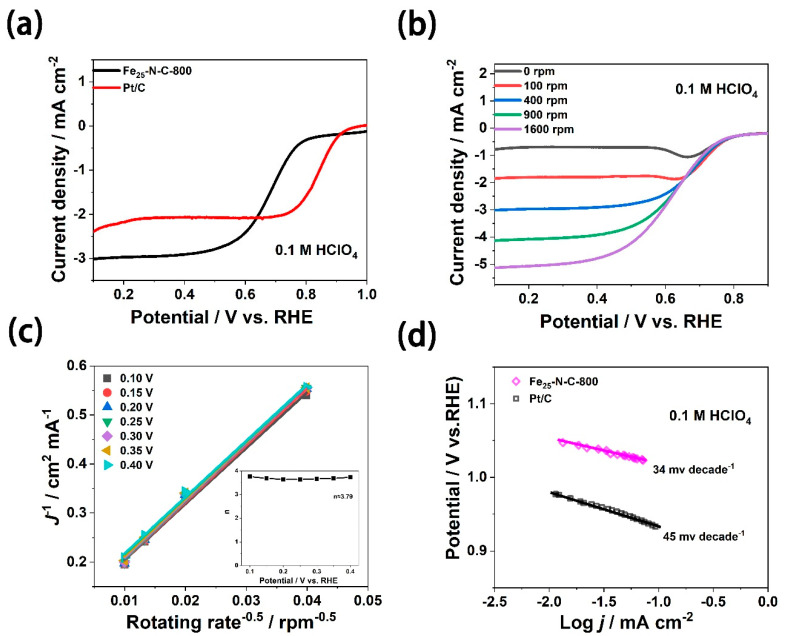
Characterizations of ORR activity in acidic medium. (**a**) RDE dates of Fe_25_-N-C-800 and Pt/C, rotation rate = 400 rpm. (**b**) Polarization curves of Fe_25_-N-C-800 at various rotation rates (increasing from top to bottom). (**c**) Koutecky–Levich plots of Fe_25_-N-C-800 and number of electrons transferred (inset) at various potentials (0.1–0.4 V vs. RHE). (**d**) Tafel slope plots of Fe_25_-N-C-800 and Pt/C derived from the LSV results.

**Figure 7 materials-13-04551-f007:**
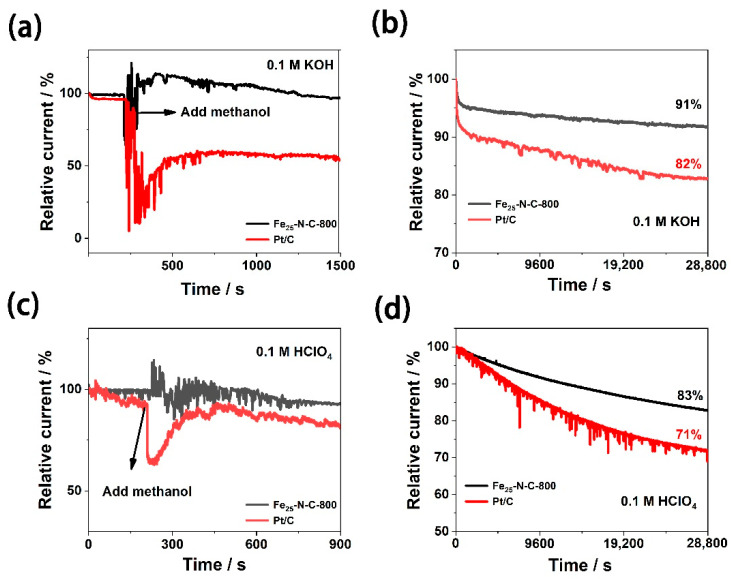
Chronoamperometric responses of Fe_25_-N-C-800 and commercial 20 wt.% Pt/C electrodes (100 rpm) upon the introduction of 3 M CH_3_OH (**a**) in 0.1 M KOH and (**c**) 0.1 M HClO_4_. Durability evaluation of Fe_25_-N-C-800 and Pt/C catalysts for 28,800 s (**b**) in 0.1 M KOH; (**d**) 0.1 M HClO_4_.

**Table 1 materials-13-04551-t001:** Elemental information of synthesized Fe_X_-N-C-800, analyzed by inductively coupled plasma (ICP).

Samples	Weight (g)	Constant Volume (mL)	Fe (wt.%)
Fe_5_-N-C-800	0.0227	10	1.18
Fe_25_-N-C-800	0.0248	10	5.35
Fe_50_-N-C-800	0.0311	10	12.65

**Table 2 materials-13-04551-t002:** X-ray photoelectron spectroscopy (XPS) element quantification of Fe_25_-N-C-Y catalysts (Y = 700, 800, 900 °C).

Samples	C%	N%	Fe%
Fe_25_-N-C-700	94.36	5.22	0.42
Fe_25_-N-C-800	94.78	4.59	0.63
Fe_25_-N-C-900	96.51	2.48	1.01

**Table 3 materials-13-04551-t003:** Electrochemical performance of different electrocatalysts for ORR.

Samples	Alkaline Electrolyte		Acidic Electrolyte		Reference
	Electron Transfer Number	Relative Current (%)/Test Time	Electron Transfer Number	Relative Current (%)/Test Time	
Fe_25_-N-C-800	3.80–3.93	91%/28,800 s	3.74–3.86	83%/28,800 s	this work
Fe-N-CNP-CNF	3.91	86.8%/20,000 s	-	-	[51]
Fe-N-MWCNTs	-	-	3.3	60%/86,400 s	[52]
Fe-N-CNFs	3.93	83.3%/20,000 s	-	-	[42]
Fe-N-C	3.8–3.9	90%/25,500 s	-	73%/25,500 s	[53]
N-doped Fe/Fe_3_C@C/RGO	3.08–3.52	91%/6000 s	-	-	[54]
Fe-N/C^800^	3.78–3.89	75.64%/7000 s	-	-	[55]

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
