# Peer review of "Facile Synthesis of the Amorphous Carbon Coated Fe-N-C Nanocatalyst with Efficient Activity for Oxygen Reduction Reaction in Acidic and Alkaline Media"

_materials, 2020, doi:10.3390/ma13204551_

Round 1
Reviewer 1 Report
Jin et al report the synthesis of Fe-N-C as ORR catalyst in acidic and alkaline media.
The manuscript need to address the following concerns prior publication
- There are a lot of reports on Fe-N-C ORR catalysts please discuss and compare the performance of the as-prepared catalyst to the literature
- The ORR performance and durability of the catalyst in acidic media is quite low, please explain
- The durability of the catalysts in alkaline and acidic medium should be benchmarked to the literature as well
- The english of the manuscript need to be polished
Reviewer 2 Report
The manuscript details a study on developing an Fe-N-C electrocatalyst for use in the oxygen reduction reaction (ORR). The electrocatalyst derives from a carbon nitride precursor doped with Fe. The effect of Fe loading and calcination temperature on electrocatalyst activity and stability is evaluated and materials are characterised by a series of standard techniques. Literature supported postulations on the nature of the active sites are provided. For the manuscript to be acceptable for publication the authors need to: (1) explicitly identify what it is that makes this work distinct from other similar published research on the same material for the ORR and (ii) provide more thorough characterisation results for their electrocatalyst variants (different Fe loadings, different calcination temperatures) so that the study adds new knowledge to the area and is less reliant on postulations supported by other literature. The preparation of Fe-N-C catalysts (including using g-C3N4) and their use for the ORR has been previously reported in a number of other publications. What is new about the research presented here? What new knowledge does it provide and what makes it distinct from other research on the same material/application? What is the purpose of including the surfactant during the synthesis? In Figure 1, what evidence do the authors provide (which is not literature based) to indicate the nature of the active site (as zoomed in on the Fe-N-C material) to justify its inclusion in the scheme? The authors use the XRD profiles to state that the Fe-based nanoparticles are Fe and Fe3C. Why is there such significant background noise for the profiles? The noise makes it difficult to discern the peaks and have confidence in the authors interpretation of the spectra, especially in relation to the presence of Fe. The key to Fe identification is the presence of the peaks at 65o (as it is distinct from the Fe3C peaks). The background noise makes it difficult to believe the authors interpretation that Fe is present in all samples. On this basis, considering the spectra, only the 50mg Fe sample appears to possess Fe due to the large spike at 44.6o. In the other spectra this peak is not readily apparent. Could it be that the Fe loading influences the potential for Fe to be present? For instance, at lower Fe loadings, Fe3C is formed in preference to Fe. At the high Fe loading there is sufficient Fe to begin forming Fe in conjunction with Fe3C. The authors need to provide better XRD patterns which allow a clear distinction between Fe and Fe3C and reconsider their interpretation of the results. The authors should provide the values of the crystal sizes they calculated from XRD. The authors state the XPS rough sweep indicates a ‘more obvious’ N signal intensity for the sample calcined at 700C. It is difficult to see this being the case, where the N signal actually looks to be less intense for this calcination temperature relative to the other temperatures. On what basis did the authors make their interpretation? TEM images (and deposit size distributions) and XPS profiles should also be provided for the samples with different Fe loadings. Same should be the case for the samples calcined at different temperatures. Additional characterisation will strengthen the work and possibly make it less reliant on simply drawing literature-based postulations. More comprehensive characterisation of the materials which have different Fe loadings and are calcined at different temperatures and linking them to the performance results will add more value to the publication than it currently holds What is the specific surface area of the materials? Characterisation of the Fe25-N-C-800 post durability testing to help understand the origin of the deactivation would also strengthen the work and should be performed. The structure of the manuscript is logical and the written English is reasonable. There are regular grammatical errors throughout the manuscript which should be removedAuthor Response
Please see the attachment.

Reviewer 3 Report
In this work, Jin and co-workers have synthesized Fe-N-C nanocatalysts based on pluronic P-123, ferric acetylacetonate and g-C3N4, and applied them as electrocatalysts for oxygen reduction reactions. This work is interesting as the obtained nanocatalysts showed a comparable activity with commercial noble-metal catalysts. Thus, this work can be useful for the development of more efficient catalysts that do not require noble metals. Given this, my opinion is that this work is publishable if the authors address the following concerns and questions:
-The title is too "strong". That is, the authors state that their nanocatalyst present "superior activity", but for starters that is really just a subjective statement, and not an objective one. Superior to what, and in what way? Furthermore, their results do not really show a superiority. The present data show instead that the developed nanocatalyst is comparable with commercial catalysts. Thus, the title must be corrected.
-What is the synthesis yield of g-C3N4?
-The authors must specify the "certain amount" of added dried melanine (page 3, line 94);
-What is the synthesis yield (mass/mass) of the Fex-N-C-Y electrocatalyst?
-In lines 228 and 229, the authors state that the content of active sites of different types of nitrogen influences the ORR performance of the catalyst. In what way? There is a particular type of nitrogen more beneficial/prejudicial to the ORR performance, and why?
-Expressions such as "most excellent" (lines 255) are not really scientific, due to being very subjective. The authors must use instead more objective expressions.
-How many replicates were performed for the experiment comparing Fe25-N-C-800 with Pt/C catalyst?
-The authors should explain better why the comparison with the Pt/C catalyst. This is the catalyst with currently the best reported performance?
-Besides comparing with Pt/C catalyst, the authors should compare their results with the results present in the literature for other available catalysts for ORR reactions.
Reviewer 4 Report
This manuscript deals with a simple synthesis strategy and characterization of a novel NMPC for the ORR in fuel cells, working in acidic and alkaline media. In the introduction, the authors justify their interest in this work, with scientific and application interest, and add 21 recent and relevant references that would help potential readers to better understand the subject involved and the procedures adopted by the authors to reach their goal. The section on materials and methods includes reagent and materials, the preparation of g-C3N4, Fex-N-C-Y electrocatalysts, working RDE, the characterization of materials and the electrochemical measurements. As in the introduction, these sections are brief, but very clear, detailing what in fact matters. An exception concerns the reference electrode; since the used reference electrode was a Ag/AgCl electrode, and all the results are given versus the RHE, the authors should show their used Evs RHE- Evs AgCl/Ag relation. In page 4, the authors should also correct part of the sentence that appears immediately after equation (3):< J,JL and Jk respectively under the specified electric potential of current density, limiting diffusion current limiting current density and dynamics>. In the subsection 3.1 on the preparation and physical characterization of the novel Fe-N-C nanocatalyst, SEM, XRD and TEM confirmed that the catalyst is made of an aggglomeration of of iron elemental and Fe3C, the iron particles being coated with amorphous carbon. HAADF-STEM confirmed the existence of iron element. The actual proportion of iron in the catalyst under different iron precursor contents was determined by ICP. XPS spectra confirmed that the catalyst basically consists of C, N, O and Fe. More information about the valence state of the atoms and surface energy distribution of the catayst were obtained by XPS. The peak obtained for Fe3C is shown to have a positive effect on ORR. Further analysis indicated that there are many chemical states of iron species in the catalyst. In sections 3.2, 3.3 the electrochmical study of the electrocatalyst was studied by RDE voltammetry (CV+LSV) and chronoamperometry. ORR activity and durability were studied in akkaline and acidic media. It was confirmed that the active site of iron speciation serves as a significant part in the process of promoting ORR electrocatalytic activity. Apart from the fact that the catalyst behaved even better than the Pt/C commercial catalyst in alkaline medium, also in acid medium it showed an excellent performance. For example, ns for alkali and acid were 3.86 and 3.79, very close to the ns for Pt/C. The Tafel plts were also very low, meaning relatively rapid reaction rate for the ORR. Excellent stability and durability was also observed. In summary, a novel active and durable catalyst , not expensive, easily synthesized, and applicable to acid and alkaline media. Paper correct, clear, well organized, scientifically deep, deserving publication.
Round 2
Reviewer 1 Report
Ok
Reviewer 2 Report
The authors have done a commendable job at addressing the original comments. The manuscript is stronger now and the novelty of the work has been clarified. Acceptable for publication.
Reviewer 3 Report
The authors have addressed all my previous comments, and so, my recommendation is for acceptance.